# Machine Learning-Based Prediction of Clinical Outcomes in Microsurgical Clipping Treatments of Cerebral Aneurysms

**DOI:** 10.3390/diagnostics14192156

**Published:** 2024-09-27

**Authors:** Corneliu Toader, Felix-Mircea Brehar, Mugurel Petrinel Radoi, Razvan-Adrian Covache-Busuioc, Luca-Andrei Glavan, Matei Grama, Antonio-Daniel Corlatescu, Horia Petre Costin, Bogdan-Gabriel Bratu, Andrei Adrian Popa, Matei Serban, Alexandru Vladimir Ciurea

**Affiliations:** 1Department of Neurosurgery “Carol Davila”, University of Medicine and Pharmacy, 030167 Bucharest, Romania; corneliu.toader@umfcd.ro (C.T.); razvan-adrian.covache-busuioc0720@stud.umfcd.ro (R.-A.C.-B.); luca-andrei.glavan0720@stud.umfcd.ro (L.-A.G.); antonio.corlatescu0920@stud.umfcd.ro (A.-D.C.); horia-petre.costin0720@stud.umfcd.ro (H.P.C.); bogdan.bratu@stud.umfcd.ro (B.-G.B.); andreiadrianpopa@stud.umfcd.ro (A.A.P.); matei.serban2021@stud.umfcd.ro (M.S.); prof.avciurea@gmail.com (A.V.C.); 2Department of Vascular Neurosurgery, National Institute of Neurology and Neurovascular Diseases, 077160 Bucharest, Romania; 3Department of Neurosurgery, Clinical Emergency Hospital “Bagdasar-Arseni”, 041915 Bucharest, Romania; 4Department of Research and Development, Syndical.io, street Icoanei 29A, 020452 Bucharest, Romania; ma-tei.grama@syndical.io; 5Neurosurgery Department, Sanador Clinical Hospital, 010991 Bucharest, Romania

**Keywords:** machine learning, ruptured intracranial aneurysm, treatment outcome, microsurgical clipping

## Abstract

**Background:** This study investigates the application of Machine Learning techniques to predict clinical outcomes in microsurgical clipping treatments of cerebral aneurysms, aiming to enhance healthcare processes through informed clinical decision making. **Methods:** Relying on a dataset of 344 patients’ preoperative characteristics, various ML classifiers were trained to predict outcomes measured by the Glasgow Outcome Scale (GOS). The study’s results were reported through the means of ROC-AUC scores for outcome prediction and the identification of key predictors using SHAP analysis. **Results:** The trained models achieved ROC-AUC scores of 0.72 ± 0.03 for specific GOS outcome prediction and 0.78 ± 0.02 for binary classification of outcomes. The SHAP explanation analysis identified intubation as the most impactful factor influencing treatment outcomes’ predictions for the trained models. **Conclusions**: The study demonstrates the potential of ML for predicting surgical outcomes of ruptured cerebral aneurysm treatments. It acknowledged the need for high-quality datasets and external validation to enhance model accuracy and generalizability.

## 1. Introduction

Medical machine learning (ML) has been subject to an exponential growth in the past decade, delivering the promise of improved healthcare delivery and clinical decision making. Traditional approaches to medical data analysis often fall short when trying to capture the subtle relationships between patient characteristics, disease pathophysiology, and treatment outcomes. In contrast, ML techniques offer a powerful framework for extracting meaningful insights from complex medical datasets, paving the way for personalized medicine and precision healthcare. Recent years have seen an alignment among scholars towards a common sense that, along with other technical and scientific advancements in the field, machine learning is going to reshape the future of neuroscience [1]. Intracranial aneurysms present a significant challenge in neurosurgery, particularly when ruptured, as they can lead to life-threatening complications such as subarachnoid hemorrhage (SAH). Understanding their formation, progression, and rupture risk is crucial for effective management and treatment. Surgical clipping remains one of the primary intervention alternatives for both ruptured and unruptured aneurysms, yet its success hinges on a multitude of patient-specific and aneurysm-related factors. In this study, our primary objective is to predict outcomes of surgical clipping interventions for prevalently ruptured aneurysms using machine learning techniques. The topic of intracranial aneurysms has been subject to multiple types of developments, with use-cases ranging from the aneurysm risk forecast [2] and prediction of periprocedural complications related to endovascular treatments [3], to outcome prediction for flow diverter treatments [4]. The general end goal of these ML developments is to improve the treatment through the means of informed decision making. These models can be integrated into clinical workflows to assist neurosurgeons in making data-driven decisions about treatment strategies for intracranial aneurysms. Given the serious implications ruptures have for the outcome of aneurysms’ surgical intervention with regard to the morbidity and mortality rates, the study of these cases becomes particularly important. This is why multiple studies have expressly focused on ruptured intracranial aneurysms and the predictive models that can be built concerning these cases, e.g., for rupture risk prediction [5,6] or the predictive factors of rerupture [7]. Our goal is to contribute to the ongoing efforts of the medical AI community by validating the transferability and applicability of existing approaches [8] within our specific scope, namely outcome predictions for surgical clipping interventions of both ruptured and unruptured intracranial aneurysms given the set of preoperative characteristics. Through this research, we aim to not only develop the predictive models but also enhance our understanding of the complex interplay between patient factors and surgical outcomes within this specific scope.

## 2. Background

In the context of outcome prediction for intracranial aneurysm treatment, the synthesis composed by Velagapudi et al. [8] categorizes the literature on this subject based on the targeted aneurysm type as unruptured, aneurysmatic subarachnoid hemorrhage (aSAH) and combined. The recent years have seen a growing interest in leveraging ML techniques for outcome prediction in the context of unruptured intracranial aneurysms (UIAs). Traditionally, the management of UIAs has relied on various clinical factors to guide treatment decisions. However, the underlying relations between these factors are of great complexity and are thus hindering the chances of accurately predicting patients’ individual outcomes. In this context, several studies have explored the feasibility of using ML models for predicting the outcomes following UIA microsurgical treatments. Stroh et al. [9] developed, for instance, an ML-based predictive model for preoperative assessment of UIA surgery outcomes, providing valuable insights into patient-specific risk factors and the potential applicability of these technologies. Similarly, Staartjes et al. [10] conducted a pilot study focusing on the development of ML models for outcome prediction in UIA surgery. Ishankulov et al. [11] also reported a good performance in predicting the outcome in the context of UIA surgical interventions. Taking note of the studies targeting ruptured and aSAH cases, Vin Shen Ban et al. [12] presented the Southwestern Aneurysm Severity Index (SASI), accurately predicting one-year functional outcomes after microsurgical clipping of ruptured aneurysms; Xia et al. [13] demonstrated that a Random Forest model can effectively predict the clinical outcome at discharge for patients with ruptured Anterior Communicating Artery aneurysms; and Zador et al. [14] used a logistic regression model with Bayesian network analysis to identify age and the World Federation of Neurosurgical Societies grade as key predictors of outcomes in acute aSAH. While these studies delivered promising results in improving outcome prediction for this pathology, there remains a need for further investigation and validation of these models across diverse patient populations and treatment settings.

## 3. Methodology

### 3.1. Dataset

The dataset was collected from 344 patients that underwent surgical interventions using the microsurgical clipping method for at least one intracranial aneurysm, predominantly saccular or berry-type intracranial aneurysms. It is composed of preoperative characteristics with an established significance in influencing the operatory outcome, such as arterial hypertension and atherosclerosis. Along with the 23 variables representing the input for the ML models, the dataset features the operatory outcome, recorded using the Glasgow Outcome Scale (GOS) for each of the individual surgical interventions. While three of the input variables (age, aneurysm diameter, and aneurysm neck) are numerical, the rest of the variables along with the GOS are categorical. The value distributions for each of those variables is presented in Table 1. A few categorical input features have missing values for an insignificant number of data points and, consequently, have been attributed a separate class to account for the missing information. While some of the input parameters represent the preexisting conditions of each individual, e.g., obesity, diabetes, others are aneurysm- and operatory-specific parameters such as the aneurysm neck and clip shape. In terms of demographics, the dataset covers a wide age range, with a higher prevalence in individuals over 30 years old and notable gender discrepancy, with more cases observed in women.

### 3.2. Model Training and Validation

The proposed ML task consists of predicting the outcome of the surgical intervention based on the preoperative parameters described earlier. This corresponds to predicting the five possible categories of GOS. The task can also be reduced to a binary classification as follows by setting GOS = 5 to correspond to a positive outcome and, conversely, GOS < 5 to a negative outcome. We conducted the experiments for both formulations and report the results in the section below.

Our experiments are constructed upon an extensive evaluation of machine learning classifiers, evaluating different state-of-the-art approaches and models, including Extra Trees (ETs), Random Forest (RF), Support Vector Machines (SVMs), K-Nearest Neighbors (KNNs), Artificial Neural Networks (ANNs), Logistic Regression (LR), Extreme Gradient Boosting (XGB), and Linear Discriminant Analysis (LDA). For each model, specific hyperparameter tuning was conducted for maximizing the outcome of the training, e.g., in the case of ETs, hyperparameters such as the number of estimators in the ensemble and the maximum depth of each tree were optimized through a grid search to enhance the model’s predictive performance (Table 2). Key libraries used for data analysis and model development included pandas for data manipulation, numpy for numerical computations, matplotlib and seaborn for data visualization, and scikit-learn for machine learning model implementation.

Techniques such as bootstrap resampling, involving iteratively sampling the dataset with replacement to estimate classifier performance variance accurately, and a stratified sampling for the train–test split are employed to ensure fairness in model evaluation. The stratified test–train split was used to address potential class imbalance issues and ensure representative sampling in both subsets, the training partition consisting of 276 data points and the test partition with 68 data points.

### 3.3. SHAP Analysis

In recent years, SHapley Additive exPlanations (SHAP) has become a significant actor in ML’s development due to its ability to provide consistent explanations that help in effective interpretations of the models [15]. Nowadays, it acts as guide for case-specific explanations, model refinement, and feature engineering by essentially quantifying each feature’s contribution to predictions. Through the means of this analysis, the input variables are ranked by impact, highlighting key predictors in the model output. In our study, we performed this analysis for the trained models using the KernelExplainer and the TreeExplainer [16] provided within the shap Python package v 0.44.1. The results for the best performing models are discussed in the following sections.

### 3.4. Ethical Standards Compliance

In line with the principles outlined in the Declaration of Helsinki, informed consent has been obtained for the healthcare data used within this study and the data handling was conducted in compliance with current GDPR regulations. All procedures performed in this study, in accordance with the ethical standards, were reviewed and approved by the Ethics Committee of the National Institute of Neurology and Neurovascular Diseases in Bucharest, Romania (IRB approval name: Ethical Review Board of National Institute of Neurology and Neurovascular Diseases; IRB number: 2/2024).

## 4. Results

Once the hyperparameter tuning was performed for the eight models included in this study, the achieved performance was assessed making use of bootstrap resampling. The results reported for the task of predicting the specific GOS outcome (Figure 1) indicate an acceptable performance in terms of ROC-AUC, with Extreme Gradient Boosting (XGB) leading with a score of 0.72 ± 0.03, also indicating relatively stable performance. Coming up next, Extra Trees (ETs) achieved a comparable ROC-AUC of 0.71 ± 0.03, followed closely by Random Forest (RF) and Support Vector Machines (SVMs) with scores of 0.70 ± 0.02 and 0.69 ± 0.02, respectively. Lower performances were observed for Artificial Neural Networks (ANNs) and Linear Discriminant Analysis (LDA), with ROC- AUC scores of 0.67 ± 0.02 and 0.67 ± 0.03. K-Nearest Neighbors (KNNs) and Logistic Regression (LR) had the lowest performances in this context, both scoring 0.65 ± 0.01 and 0.65 ± 0.03, respectively.

When reducing the task to the binary classification distinguishing between the positive outcome, corresponding to GOS = 5, and the negative outcomes, for GOS < 5, a different model ranking and, as Figure 1 shows in terms of the outcome prediction performance of the different models measured by means of ROC-AUC, a significant increase in the performances were established, with the highest ROC-AUC of 0.78 ± 0.02retained. being achieved this time by the ET classifier. Following closely behind, KNN achieved a notable ROC-AUC score of 0.75 ± 0.03. LR and RF also performed well, both attaining ROC-AUC scores of 0.74 ± 0.04 and 0.74 ± 0.03, respectively. LDA achieved a moderate performance with a score of 0.72 ± 0.04. ANN, SVM, and XGB all showed similar performances, each obtaining ROC-AUC scores of 0.70 ± 0.04, ± 0.04, and ± 0.03, respectively. The SHAP analysis we conducted for the best performing models (ET, ANN, XGB, and RF) pointed out the most influential factors affecting the outcome for each of the models (Figure 2). The most notable finding that emerged is the consistent significance of intubation in predicting the GOS value across all models. This feature shows up as the highest-ranking factor in terms of importance. Another interesting observation comes in regard to vasospasm, which consistently shows up among the three most influential parameters. The Hunt and Hess Scale has also consistently ranked high for three out of four of the best-performing models. The observations resulting from our experiments are further discussed in the following section.

## 5. Discussion

In our study, the application of machine learning (ML) models to predict clinical outcomes of microsurgical clipping for cerebral aneurysms shows promising results, achieving good and acceptable performance levels. These findings align with previous research, highlighting the potential of ML to enhance clinical decision making and patient care in neurosurgery. In a similar study conducted by Stroh et al. within the scope of unruptured cerebral aneurysms and the use of ML models for predicting clinical outcomes for microsurgical clipping [9], a good performance for the trained ML models has been reported, indicating a significant potential to enhance prognostic accuracy and customize surgical approaches for individual patients. In their case, for the task of predicting the positive post-operative outcome in terms of GOS, a slightly higher ROC-AUC value has been reported over the test set and lower ROC-AUC for the external validation, 0.79 ± 0.07 and 0.62 ± 0.02, respectively. The models showcased good performance for differentiating the outcomes based on preoperative indicators and clinical parameters. This sort of prognosis can play a part in enhancing nuanced risk stratification and personalized treatment planning, resonating with the ongoing shift towards precision medicine in neurosurgery. However, challenges in improving the generalizability and validity of these predictive models across diverse healthcare settings were noted, highlighted by variations in model performance during external validation. To address these issues, future efforts will need to focus on integrating larger, multi-institutional datasets and continually updating algorithms to reflect current clinical practices, ensuring that ML tools effectively support clinical decision making in the management of cerebral aneurysms. By conducting the training phase across multiple regional datasets, the models’ performance can improve and thus increase their applicability in real-time prediction of ruptured aneurysms, enabling an enhanced tracking of the patients’ evolution for this pathology. Correctly assessing the outcomes of unstable patients can lead to correct therapeutic conducts, which in turn will lead to better overall care and survival. Moreover, an accurate model can be a helpful tool for neurosurgeons in determining the urgency of surgical interventions and for intensive care specialists to formulate a better image of the outcome of each case.

One of the core strengths of this study lies in its attempt to model clinical outcomes in a real-world setting, where both ruptured and unruptured aneurysms have been routinely encountered. While the majority of the cases in the dataset are ruptured aneurysms, the inclusion of unruptured aneurysms provides a more accurate representation of the treated population in clinical practice subject to this study. In this context, both ruptured and unruptured aneurysms are treated, often within the same clinical environment. Therefore, by including both classes, the model performance is measured over the full spectrum of cases, enabling the findings to be more widely applicable to clinical decision making. Not including both ruptured and unruptured aneurysms would potentially limit its utility in this specific clinical context, and in fact, validating the model on a heterogeneous dataset that includes both types of cases is essential to its generalizability, particularly as ML techniques are applied to increasingly complex patient populations. This aspect is critical for the future development of a versatile and adaptable tool that performs well over the diverse range of neurosurgery cases.

Our study identified several key parameters as being highly influential in predicting the clinical outcomes of microsurgical clipping for cerebral aneurysms, through the means of modern techniques such as explainable machine learning. These relevant parameters, determined through SHAP analysis, may provide valuable insights into the factors that most significantly impact the model prediction outcomes. One of the relevant factors was intubation, as it consistently showed up as the most impactful parameter across the best performing models. The requirement for intubation is often associated with more severe cases and can indicate a higher risk of complications or poorer baseline patient status. Another important parameter was the vasospasm, which is particularly critical as it is associated with delayed cerebral ischemia, a common and severe complication following aneurysmal subarachnoid hemorrhage. Its early prediction can influence treatment decisions and potentially improve outcomes. The Hunt and Hess Scale assesses the initial clinical condition of patients with subarachnoid hemorrhage, and was also highlighted as a key predictor. It reflects the severity of the hemorrhage and the patient’s neurological status, which are crucial for outcome prediction. These findings underscore the complexity of cerebral aneurysm management and the value of ML in capturing and analyzing these nuances.

Through the means of such studies, we can explore intricate patterns that are often inaccessible through traditional statistical methods, particularly when dealing with high-dimensional datasets, as demonstrated by the SHAP analysis. For example, the identification of intubation as the most impactful feature was consistent across all models, suggesting a non-obvious link between the preoperative conditions in question and post-surgical outcomes. This level of interpretability allows ML models to highlight parameters previously underappreciated and help refine the understanding of which factors significantly influence patient outcomes in different settings, beyond what clinicians may intuitively assess.

### Limitations and Future Directions

The present study encountered several limitations that should inform future directions of research regarding this subject. The dataset displayed imbalances across certain features, which could skew model performance. When low performance levels were observed, this may be an indication of inherent deficiencies within the data, suggesting a need for higher quality training sets to improve model efficacy. In any ML-based analysis, both the data and model selection play crucial roles in determining the outcome prediction. In this study, while models such as Extreme Gradient Boosting (XGB) and Extra Trees (ETs) performed well, the dataset itself introduces limitations that may have influenced these results. For example, the presence of only three basilar aneurysm cases and certain missing points could skew predictions. When there is an imbalance in such critical features, models tend to overfit to the more frequent data points, reducing their ability to generalize to rare events. Nevertheless, in healthcare, the imbalanced nature of data reflects the actual prevalence of pathological characteristics in the population. Manipulating this imbalance, e.g., oversampling rare cases or undersampling common ones, can lead to models that do not generalize well to real-world scenarios due to the sampling biases that might be introduced by the application of such techniques. Handling missing data points can also significantly affect model robustness. Making use of techniques such as imputing missing data by creating separate classes or applying methods like mean imputation that ensure no loss of data may also lead to the same adverse effect of oversimplifying underlying patterns. In many clinical settings, predictions that reflect the true incidence rates of diseases are more meaningful for decision making. Manipulating the data can reduce the utility of a model for clinicians by distorting the probability estimates.

As healthcare moves towards more nuanced outcome measures beyond single-dimensional scales like GOS, current ML models may face limitations. More sophisticated measures could involve a comprehensive view that would combine physical, cognitive, and psychological recovery indicators. Furthermore, the dataset’s single origin raises concerns about potential biases and disparities, underscoring the importance of multi-institutional, multi-regional datasets to mitigate these biases and enhance the reliability. Moving forward, future research should prioritize standardizing methodologies, validating findings across varied cohorts, and exploring innovative approaches to enhance both the accuracy and generalizability. Exploring additional predictive features, such as genetic markers and advanced imaging parameters, could further enhance the accuracy and utility of these models. The integration of these predictive models into clinical practice could significantly enhance decision-making processes, potentially leading to better patient outcomes and more personalized treatment plans.

## 6. Conclusions

Our study shows that the ML models trained for predicting clinical outcomes of microsurgical clipping treatments for intracranial aneurysms achieved acceptable performance levels when trained using the collected patient data. These results come in alignment with similar studies’ conclusions reported in the literature, indicating that the use of ML techniques under this scope presents a high potential for enhancing clinical decision making. However, before these models can be deployed in real-world systems with the required levels of generalizability and reliability, further research focusing on diversified patient populations is needed for validation and further refinements. Within the domain of neurosurgery, machine learning holds great potential of enhancing the management of intracranial aneurysms. These vascular anomalies present a formidable challenge to neurosurgeons due to their unpredictable nature and potentially devastating consequences. The development of decision support systems incorporating these predictive models could provide real-time risk assessments and personalized treatment recommendations. Overall, the results contribute to the growing body of evidence supporting the integration of ML techniques into healthcare for better patient management and treatment outcomes.

## Figures and Tables

**Figure 1 diagnostics-14-02156-f001:**
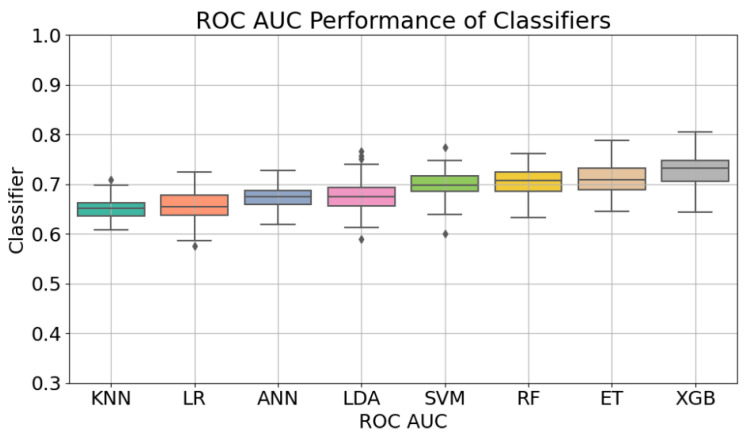
Outcome prediction performance of the different models measured by ROC-AUC.

**Figure 2 diagnostics-14-02156-f002:**
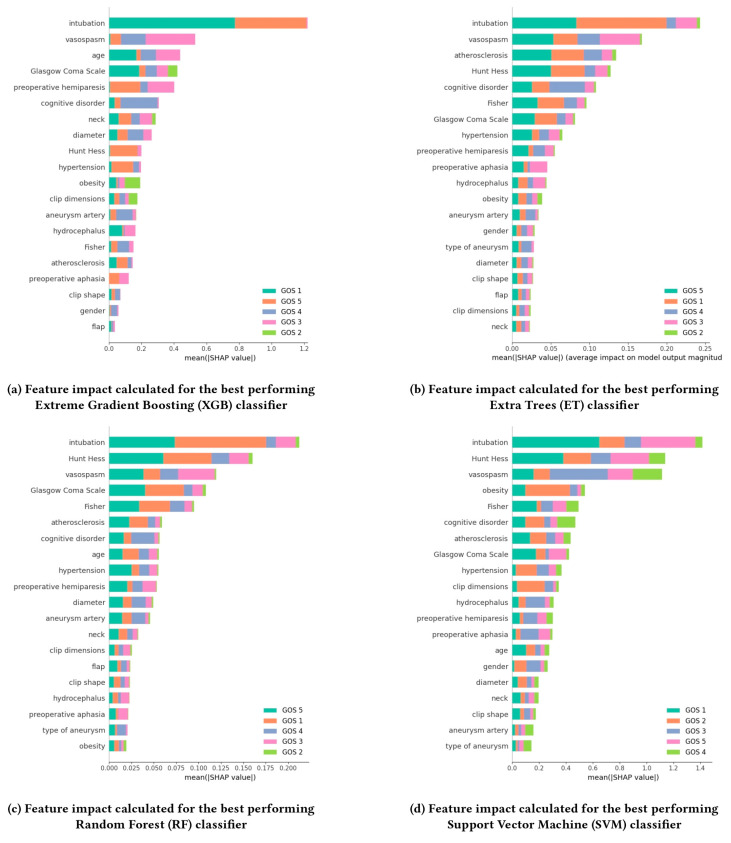
SHAP analysis for the four best performing models (XGB, ET, RF, SVM) consisting of the feature ranking based on the average impact on model output magnitude, with higher values corresponding to a higher impact on GOS prediction.

**Table 1 diagnostics-14-02156-t001:** Characteristics of the variables recorded for the study cohort (no. of cases n = 344).

Variable	Description
Age (years), mean ± σ	55.05 ± 11.45
Female gender	219 (63.7%)
Aneurysm Diameter (mm), mean ± σ	7.36 ± 3.51
Aneurysm Neck, mean ± σ	3.93 ± 1.64
Clip Shape, *n* (%)	
Straight	201 (58.4%)
Curved	102 (29.7%)
Angular	21 (6.1%)
Wrap	17 (4.9%)
Arterial Hypertension, *n* (%)	
Grade II	145 (42.2%)
Grade I	67 (19.5%)
Grade III	8 (2.3%)
Atherosclerosis	108 (31.4%)
Diabetes Mellitus (Type 2)	17 (4.9%)
Obesity, *n* (%)	
Not present	256 (74.4%)
Grade 2	53 (15.4%)
Morbid	30 (8.7%)
Grade 1	2 (0.6%)
Preoperative Aphasia	9 (2.6%)
Cognitive Disorder	118 (34.3%)
Intubation	76 (22.1%)
Hemorrhage	311 (90.4%)
Hydrocephalus	124 (36.0%)
Ruptured	308 (89.5%)
Aneurysm Artery, *n* (%)	
A-Comm	152 (44.2%)
MCA	93 (27.0%)
PCA	35 (10.2%)
Pericallosal	13 (3.8%)
P-Comm	11 (3.2%)
Ophthalmic	11 (3.2%)
Choroidal	7 (2.0%)
ICA	6 (1.7%)
Fetal P-Comm	5 (1.4%)
Basilar	3 (0.9%)
Other (*)	8 (2.3%)
Craniotomy, *n* (%)	
Pterional	178 (51.7%)
Frontobasal	137 (39.8%)
Frontobasal Paramedian	16 (4.7%)
Other	11 (3.2%)
Suboccipital	2 (0.6%)
Vasospasm	101 (29.4%)
Glasgow Coma Scale (GCS), *n* (%)	
GCS 14	219 (63.7%)
GCS 13	57 (16.6%)
GCS 7	33 (9.6%)
GCS 15	29 (8.4%)
GCS 9	18 (5.2%)
GCS 8	16 (4.7%)
GCS 12	15 (4.4%)
Other	59 (17.1%)
Preoperative Hemiparesis, *n* (%)	
5	233 (67.4%)
2	63 (18.3%)
3	26 (7.6%)
1	10 (2.9%)
4	10 (2.9%)
Other	3 (0.9%)
Fisher Scale, *n* (%)	
Grade 3	142 (41.0%)
Grade 4	98 (28.5%)
Grade 2	72 (20.9%)
Grade 1	31 (9.0%)
Grade 0	2 (0.6%)
Glasgow Outcome Scale (GOS), *n* (%)	
GOS 5	141 (41.0%)
GOS 1	77 (22.4%)
GOS 4	71 (20.6%)
GOS 3	50 (14.5)
GOS 2	5 (1.5%)
σ—standard deviation;	
A-Comm—Anterior Communicating Artery;	
MCA—Middle Cerebral Artery;	
PCA—Posterior Cerebral Artery;	
P-Comm—Posterior Communicating Artery;	
ICA—Internal Carotid Artery;	
Fetal P-Comm—Fetal Posterior Communicating Artery;	
(*) including Carotid Cave, Hypophyseal Artery, Distal Anterior Cerebral Artery, Internal Carotid Artery bifurcation, Posterior Inferior Cerebellar Artery, Vertebral Artery, Anterior Cerebral Artery.	

**Table 2 diagnostics-14-02156-t002:** Hyperparameter tuning settings for each classifier.

Classifier	Hyperparameters
ANN	hidden_layer_sizes: [(50, 50), (100, 100)], alpha: [0.0001, 0.001, 0.01]
SVM	C: [0.1, 1, 10], kernel: [‘linear’, ‘rbf’], gamma: [’scale’, ’auto’]
XGB	learning_rate: [0.01, 0.1, 0.2], n_estimators: [100, 200, 300]
RF	n_estimators: [50, 100, 200], max_depth: [None, 10, 20], min_samples_split: [2, 5, 10]
KNN	n_neighbors: [3, 5, 7], weights: [‘uniform’, ‘distance’]
LR	C: [0.1, 1, 10], penalty: [‘l2’, ‘none’]LDA solver: [‘svd’, ‘lsqr’, ‘eigen’]

## Data Availability

Data requests should be performed according to GDPR laws.

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
