# Peer review of "Machine Learning-Based Prediction of Clinical Outcomes in Microsurgical Clipping Treatments of Cerebral Aneurysms"

_diagnostics, 2024, doi:10.3390/diagnostics14192156_

Round 1

Reviewer 1 Report

Comments and Suggestions for Authors

The paper is well written and clear in its findings. The author do a good job in describing the models they created and using a SHAP analysis to outline underlying significant individual factors. Certainly one strength the variable models used That is very helpful to the readers. However, I think one major draw back of this publication is the mixing of ruptured and unruptured aneurysms. The vast majority of the aneurysms treated were ruptured. Ultimately, my concern is that including the unruptured aneurysms will ultimately weaken the model and by focusing on ruptured aneurysms the findings would be stronger. Especially given that the majority of findings in the SHAP analysis are associated with only ruptured aneurysms. 

Author Response

Dear Reviewer

We have addressed every concern accordingly. 

Thank you for reviewing our manuscript

Reviewer 2 Report

Comments and Suggestions for Authors

The authors analysed clinical parameters at presentation and preoperative factors for 344 cases of clipping of ruptured intracranial aneurysms using machine learning to predict outcome. Their methods  and analysis were thorough and they concluded that ML showed acceptable performance using  ROC-AUC with Extreme Gradient Boosting to potentially predict outcome. 

The importance of this study that it has identified and highlighted [1] the need for high quality datasets used for ML and [2] the need to externally validate the data and perform studies in different clinical settings to enhance the model accuracy.

In general before applying ML to clinical setting such as outcome after clipping ruptured aneurysms, an area of deficiency or requirement of improvement that is needed to be identified, then present ML with a specific question, identify results that we did not already know, try to explain deviations in findings from existing knowledge and finally assess whether the findings would have a useful impact on clinical practise. 

As such the authors' quest was to enhance clinical decision making and patient care for specific individual patients. The data included, some with numerical, some categorical and some had missing values. In their analysis, the authors evaluated through ML included different approaches and models (ET, RF, SVM, KNN, ANN, XGB, LDA). The results regarding the order of the factors influencing outcome were presented for the four best performing models using SHAP analysis. The outcome assessment was the GOS. 

Although the models agreed in identifying intubation as the top ranking factor and both vasospasm and grade as important factors, however there were variations in the ranking of other factors (such as age) between the different models. 

Given that the aim is to identify factors that are specific for individual patients and recommendation that future analyses to be performed in future studies in different setting, it will enhance the manuscript and clarify the results if the authors are able to express and opinion regarding these variations especially to adopt the best models in future studies. For example how much did the data factors (such as number of cases, limited factors e.g. only 3 were basilar aneurysms, the range, the missing points, ....etc.) vs. the choice of the model (dealing with different data categories, dealing with missing points, ...... etc.) played a predominant role or can an explanation regarding potential variations in results. If possible a short comment as to how ML has revealed facts that are not already known and how can it determine parameters that are not important. What is the potential deficiencies of different models to cope if multiple more sophisticated outcome measure are to be applied beyond just a scale such as the GOS. 

  1.  
    1.  
  1.  

    1.  
    2.  
  1.  

Author Response

(The authors gave the same response as above.)
